# The Risk of Hospitalization in COVID-19 Patients Can Be Predicted by Lung Ultrasound in Primary Care

**DOI:** 10.3390/ijerph18116083

**Published:** 2021-06-04

**Authors:** Javier Martínez-Redondo, Carles Comas, Jesús Pujol Salud, Montserrat Crespo-Pons, Cristina García-Serrano, Marta Ortega Bravo, Jose María Palacín Peruga

**Affiliations:** 1Balaguer Primary Care Center, Institut Català de la Salut (ICS), 25600 Lleida, Spain; jmartinez.lleida.ics@gencat.cat (J.M.-R.); jpujol.lleida.ics@gencat.cat (J.P.S.); mcrespo.lleida.ics@gencat.cat (M.C.-P.); cgarcia.lleida.ics@gencat.cat (C.G.-S.); 2Department of Mathematics, Campus Cappont, University of Lleida, 25001 Lleida, Spain; carles.comas@udl.cat; 3Biomedical Research Institute (IRB Lleida), Universitat de Lleida (UdL), 25198 Lleida, Spain; 4Research Group in Therapies in Primary Care (GRETAPS), 25007 Lleida, Spain; 5Research Support Unit Lleida, Fundació Institut Universitari per a la Recerca a l’Atenció Primària de Salut Jordi Gol i Gurina (IDIAPJGol), 25007 Lleida, Spain; 6Onze de Setembre Primary Care Center, Institut Català de la Salut (ICS), Passeig Onze de Setembre, 25005 Lleida, Spain

**Keywords:** primary health-care, hospitalization, assessment risk, ultrasound, lung interstitial disease, COVID-19

## Abstract

Background: The usefulness of Lung Ultrasound (LUS) for the diagnosis of interstitial syndrome caused by COVID-19 has been broadly described. The aim of this study was to evaluate if LUS may predict the complications (hospital admission) of COVID-19 pneumonia in primary care patients. Methods: This observational study collects data from a cohort of 279 patients with clinical symptoms of COVID-19 pneumonia who attended the Balaguer Primary Health Care Area between 16 March 2020 and 30 September 2020. We collected the results of LUS scans reported by one general practitioner. We created a database and analysed the absolute and relative frequencies of LUS findings and their association with hospital admission. We found that different LUS patterns (diffuse, attenuated diffuse, and predominantly unilateral) were risk factors for hospital admission (*p* < 0.05). Additionally, an evolutionary pattern during the acute phase represented a risk factor (*p* = 0.0019). On the contrary, a normal ultrasound pattern was a protective factor (*p* = 0.0037). Finally, the presence of focal interstitial pattern was not associated with hospital admission (*p* = 0.4918). Conclusion: The lung ultrasound was useful to predict complications in COVID-19 pneumonia and to diagnose other lung diseases such as cancer, tuberculosis, pulmonary embolism, chronic interstitial pneumopathy, pleuropericarditis, pneumonia or heart failure.

## 1. Introduction

The recent COVID-19 virus emerged in China in December 2019. On 11 March 2020, the World Health Organization declared COVID-19 a pandemic [1]. By that date, the pandemic began to notably affect Spain, with 2128 cases and 47 deaths [2], which increased rapidly to 94,417 cases and 8189 deaths by 31 March 2020 [3]. As of 14 May 2021, the cumulative incidence in Spain is 3,604,799 cases and 79,339 deaths. 

During the month of February 2020, different studies reported that the most frequent complication in COVID-19 infection was interstitial pneumonia [4,5,6]. Likewise, during the months of February and March 2020, some publications pointed out the usefulness of lung ultrasound (LUS) to diagnose interstitial syndrome caused by COVID-19 [7,8]. Indeed, already in 2012, Volpicelli et al. provided scientific evidence that lung ultrasound can diagnose lung interstitial syndrome [9].

Different reasons led doctors in the Lleida territory (Spain) to use ultrasound as a fundamental tool to face the challenge of early diagnosis of COVID-19 pneumonia: the lack of RT-PCR tests, the difficult access to computerized tomography, and the rapid evolution of the disease. The purpose was to identify patients with pulmonary complications (early diagnosis) and avoid referrals to the hospital that did not end up being admitted. Specifically, in the rural Balaguer Primary Health Care Area, we started to use LUS in March 2020.

To date, the scientific community has accepted that LUS is a first-line tool in the early diagnosis of COVID-19 pneumonia [10]. Some studies assessed the ability of LUS to predict the evolution of the patient [11,12,13,14,15,16], however, most of them were carried out in hospital settings; therefore, they did not consider situations prior to hospital admission. Few studies addressed the use of LUS in primary health care (PHC) [17,18]. As far as we know, there is one publication only regarding the utility of LUS in PHC in order to make a decision about a patient’s hospital admission [18].

The aim of this study was to evaluate the association of certain patterns of LUS and other LUS findings with hospital admission in primary care patients with confirmed COVID-19 pneumonia. We also intended to describe other LUS findings in patients with suspected and not confirmed COVID-19 pneumonia, as well as to calculate the percentage of patients with ultrasound diagnosis of COVID-19 pneumonia confirmed by a biological test, who did not require hospitalization and were managed at the primary care service.

## 2. Materials and Methods

### 2.1. Study Design and Participants

This observational descriptive study was carried out at the Balaguer Primary Care Centre (Balaguer, Spain). We retrospectively analysed a consecutive sample of patients admitted to the emergency room of this primary care centre between 16 March 2020 and 30 September 2020. The inclusion criteria were: (a) older than 18 years; (b) availability of ultrasound pulmonary imaging; (c) confirmed or suspected COVID-19 infection defined by the existence of, at least, one of the symptoms compatible with COVID-19 pneumonia (cough, dyspnoea, fever as temperature >37 °C, oxygen saturation <95%); (d) availability of biological tests (confirmation of SARS-CoV2 infection with positive RT-PCR, serology, or rapid antigen test; negative RT-PCR was the only biological test used to rule out SARS-CoV-2 infection); and (e) consent to participate in the study. The only exclusion criterion was previous admission to the hospital for COVID-19-related complications.

We collected patients’ data from electronic medical records: demographic variables (age, sex); clinical variables (evolution throughout the acute phase, symptoms); analytical variables (laboratory blood tests, PCR test from nasopharyngeal swab); and results of the LUS exploration by a general practitioner (GP).

### 2.2. Imaging Protocol and Interpretation

LUS was performed in the patient’s back in a seated position, avoiding a face-to-face approach for the examiner. We scanned four areas (craniocaudal lines: parallel and 5 cm from the thoracic spine, medial border of scapula, posterior axillary line, medial axillary line) per lung. We used a convex probe CH5-2 (linear probe VF10-5 for very thin patients) with a band of 2–5 MHz (5–10 MHz for linear probe) width on a Siemens Acuson X150 system (Siemens S.A., Seoul, Korea). The exam was performed using B mode with a general setting (abdominal) focused at pleural line level (1–3 cm underneath in case of obese patients) and a depth set to 6–12 cm.

LUS results were interpreted by the same GP, who had more than 3 years of experience in ultrasound scanning. The images obtained were classified according to the arrangement of the B-lines (patterns) following the recommendations of Volpicelli et al. [9], and the finding of other patterns not described in these recommendations.

Patients with suspected pneumonia were followed by LUS throughout the acute phase. In the current study we found patients with evolutionary patterns. For the hospitalization risk analysis we only considered the most evolved LUS pattern (greater number of B lines) that the patient had reached during the course of the disease.

### 2.3. Ethical Approval

This study was approved by the Ethical Committee of Institut d’Investigació en Atenció Primària Jordi Gol i Gurina (Barcelona, Spain) (registration number p20/138).

### 2.4. Statistical Analysis

We measured the absolute and relative frequencies of LUS findings in patients with confirmed RT-PCR with COVID-19 pneumonia. We studied the dispersion of the sample indicating the interquartile ranges and performed t-test analysis to compare means of time between the appearance of symptoms and the appearance of the different ultrasound patterns. Chi-square analysis (X2) was used to assess the association between LUS findings and hospital admission and *p* < 0.05 was considered statistically significant in all tests. Statistical analysis was performed using the R statistical package, version 4.0.2. (https://www.r-project.org/index.html, accessed on 20 January 2021).

## 3. Results

### 3.1. Patient Demographics and Analysis of Symptoms

A total of 279 patients were included in this study with a mean age of 52.95 years, from whom 142 (50.89%) were men. Patients’ distribution is shown in Figure 1.

A total of 172 (61.65%) patients presented a pathological LUS. Of them, 131(76.17%) had a positive RT-PCR and were diagnosed with COVID-19 pneumonia. The other 41(23.83%) had a negative RT-PCR and their analysis will be described further on.

107 (38.35%) patients presented a normal LUS. Of them, 30 (28.04%) had a positive RT-PCR and were diagnosed with COVID-19 infection without pulmonary involvement. The other 77(71.96%) had a normal LUS and a negative RT-PCR.

#### 3.1.1. Symptom Distribution

The final number of patients with LUS examination and a positive RT-PCR result was 161.The most frequent symptoms were fever (132, 81.98%), cough (47, 29.19%), dyspnoea (45, 27.95%), and SaO2 < 95% (31, 19.25%).

#### 3.1.2. Ultrasound Patterns Observed According to the Presence and Arrangement of the B Lines in Patients with Positive RT-PCR

We found different COVID-19 interstitial patterns that consensually we called: diffuse (bilateral pneumonia); attenuated diffuse (bilateral pneumonia); unilateral predominant (unilateral pneumonia); focal (focal pneumonia); and normal (without lung involvement). In Table 1, patterns description and number of cases are shown.

In Figure 2, the difference between normal and pathological lung field in longitudinal section between two ribs in patients with COVID-19 pneumonia is shown.

The average days between the onset of symptoms and the LUS findings were 6.59 (IQR 5), 7.33 (IQR 4.5), 4.82 (IQR 3), 7.16 (IQR 4.25) and 3.56 (IQR 3.75), for diffuse, attenuated diffuse, predominantly unilateral, focal, and normal patterns, respectively.

We observed significant differences in the average number of days from the onset of the symptoms to the LUS examination between the diffuse and normal patterns (*p* = 0.0032) and between diffuse and predominantly unilateral patterns (*p* = 0.0002) as well as between the focal and normal patterns (*p* = 0.0207).

#### 3.1.3. Evolution of LUS Patterns in Patients with Positive RT-PCR

From the onset of symptoms to the stabilization of the disease, 11 patients with COVID-19 pneumonia presented an evolution of the ultrasound pattern to diffuse: one patient with an attenuated diffuse pattern, eight patients with a unilateral predominant interstitial pattern, one with a focal pattern, and one with a normal pattern.

In our study, all patients were followed during the acute phase of the disease and 100% of patients with confirmed COVID-19 pneumonia who were not admitted to hospital presented clinical recovery, either before or during the third week from the onset of symptoms.

#### 3.1.4. Other Ultrasound Findings in Patients with Positive RT-PCR

Apart from the distribution of the B-lines, we observed other aspects in pathological LUSs: coalescence of B-lines, diaphragm in rising sun, irregular and/or thickened pleura, decreased sliding, consolidations, and pleural effusion. These were more frequent in the diffuse interstitial pattern.

In Table 2, we show the distribution of these LUS findings according to the different patterns.

In Figure 3, images of other ultrasound findings found in patients with COVID-19 pneumonia confirmed by RT-PCR are shown.

#### 3.1.5. LUS Findings as Either Risk or Protector Factors, for Hospital Admission of Patients with Positive RT-PCR

In patients with LUS examination and a positive RT-PCR result (“*n*” = 161), a normal ultrasound pattern was a protective factor for hospital admission (X2 = 8.4005, *p* = 0.0037). Specifically, diffuse, attenuated diffuse, and predominantly unilateral patterns represented a risk factor for hospitalization (X2 = 9.9598, *p* = 0.0016; X2 = 13.236, *p* = 0.0002; X2 = 4.2480, *p* = 0.0393, respectively). We did not observe a correlation of focal interstitial pattern with hospital admission (X2 = 0.4725, *p* = 0.4918). Finally, an evolutionary pattern of pneumonia was also a risk factor (X2 = 7.4217, *p* = 0.0064).

In patients with confirmed COVID-19 pneumonia and diffuse interstitial pattern (the most frequent pathological ultrasound pattern, and the one in which the greatest number of other ultrasound findings have been found), no correlation was seen in the variables: coalescence of B lines, diaphragm in rising sun, pleural irregularity, decreased sliding, consolidation, and pleural effusion, with hospital admission (X2 = 0.2527, *p* = 0.6151; X2 = 3.0583, *p* = 0.0803, X2 = 1.9544, *p* = 0.1621, X2 = 0.2977, *p* = 0.5853, X2 = 0.1655, *p* = 0.6841; andX2 = 0.0018, *p* = 0.9654, respectively).

In our study, 37 of the 131 (22.24%) patients with an ultrasound diagnosis of COVID-19 pneumonia confirmed, who also had severe dyspnoea and SaO_2_ < 95%, were admitted to the hospital because of poor clinical evolution; therefore, 71.75% of the patients could have been managed by the primary care service.

In Table 3, we show the distribution and association between hospital admissions and the different pathological LUS patterns.

#### 3.1.6. Distribution of Other Diagnoses According LUS Pattern in Patients with Negative RT-PCR

We observed 41 patients with pathological LUS and negative RT-PCR.

14 of them showed a diffuse interstitial pattern. Among them, three were finally diagnosed with pneumonia (great consolidation with air bronchogram); one with pulmonary cancer; one with pleuropericarditis; one with pulmonary tuberculosis; one with chronic interstitial pneumopathy; four patients presented several fields with more than three B lines with homogeneous distribution in the lung bases and without other pathological ultrasound findings and were hence oriented as congestive heart failure. There were three patients in whom we could not establish the diagnosis but presented good evolution.

One of the patients showed an attenuated diffuse interstitial pattern so we could not establish the diagnosis. He eventually presented a good evolution.

13 patients had a predominantly unilateral interstitial pattern. Among them, one was finally diagnosed with lymphoma, and one with heart failure. In 11 patients we could not establish the diagnosis and presented good evolution.

12 patients had a local interstitial pattern. Among them, one was finally diagnosed with a pulmonary embolism, one with emphysema, one with heart failure. In 9 patients we could not establish the diagnosis and presented good evolution.

One patient presented a pathological ultrasound with a big consolidation and absence of B lines and presented a negative RT-PCR. He was diagnosed with pneumonia.

## 4. Discussion

### 4.1. Summary

During the initial (acute) phase of the disease, we were able to establish a classification of several ultrasound patterns for COVID-19 pneumonia on the basis of the distribution of B-lines: diffuse interstitial, attenuated diffuse interstitial, unilateral predominant, focal, and normal. We were unable to establish a clear temporal relationship between the onset of symptoms and the ultrasound pattern presented by the patients. We consider that the response is multifactorial, and it depends on the baseline situation of the patient, the initial viral load, the subjectivity in which patients perceive their disease, etc. A normal ultrasound pattern in patients with positive RT-PCR was a protective factor for hospital admission. On the contrary, evolutionary patterns, or diffuse, attenuated diffuse, and unilateral predominant interstitial patterns represented risk factors for hospitalization. The presence of a focal interstitial pattern, coalescence of B lines, diaphragm in rising sun, pleural irregularity, decreased sliding, consolidation, and pleural effusion were not correlated with hospital admission.

In our study, 71.75% of patients with an ultrasound diagnosis of COVID-19 pneumonia confirmed by RT-PCR could be controlled in our primary care centre and did not require hospital admission. All the patients who were not admitted to the hospital presented clinical recovery before or during the third week from the onset of the symptoms.

For these reasons, we consider that LUS is a very useful tool to decide the type of follow-up that should be carried out in patients with COVID-19 pneumonia until recovery or hospital referral, but in no case should it be the only criterion to decide referral or hospital admission.

LUS has also been a key diagnostic tool to detect other lung pathologies (e.g., tuberculosis, lung cancer, pneumonia) that presented symptoms similar to COVID-19 pneumonia.

Finally, in our study, patients most frequently presented fever, followed by cough, as reported in other studies [4,5,17,19]. The LUS findings of our study are similar to those shown in other studies [7,8,9,10,11,12,13,14,18].

### 4.2. Strengths and Limitations

The most important strength of our study is that all ultrasound examinations were performed by the same examiner. Ultrasound is an operator-dependent technique, and, therefore, the presence of multiple analysts and scanners can lead to different interpretations. One limitation is the retrospective nature of the study that carries inherent bias.

In our sample, the number of patients with attenuated diffuse pattern was very small, so we understand that the results that refer to this pattern can be questioned. 

Moreover, until mid-May 2020, RT-PCR tests were not available in the primary care centre where this study was carried out; however, patients were referred to the hospital, where professionals followed the same protocol for sample collection.

LUS can detect patterns but cannot detect the diseases that cause them. For this reason, it could be the case that the pathological LUS pattern could be a consequence of a previous pulmonary pathology, despite presenting positive RT-PCR.

Finally, our sample size was relatively small. Nevertheless, it was much larger than other published studies [7,8,9,10,11,12,13,14,18,20,21,22] and, to our knowledge, it is the largest sample examined by a single explorer. In conclusion, we consider that more studies should be carried out, so the results can finally be used by the scientific community.

### 4.3. Comparison with Existing Literature

Several studies investigated the prognostic value of ultrasound findings in patients with COVID-19 pneumonia; however, most of them were carried out in patients admitted to hospitals and, therefore, in a more advanced stage of the disease [11,12,13,14,15,16]. Many of these studies used pleural abnormalities or the presence of condensations to categorize the risk of deterioration. However, the meta-analysis by Mohamed et al. [14] already concluded that findings other than B pattern were very heterogeneous and depended on various factors.

We are only aware of one other research with a similar aim carried out in primary care; the study by Calvo-Cebrian et al. [18]. However, this project presents profound differences from ours: only patients with clinical suspicion of COVID-19 pneumonia were studied, as RT-PCR tests were not available; the “*n*” was lower; the ultrasound scans were performed by several explorers; and other ultrasound findings outside the B-lines were included in the severity scale. Our study proposes a simpler classification, based on ultrasound patterns according to the arrangement of B-lines, leaving aside other ultrasound findings. The novelty introduced by this study is that this classification based on the presence of protective or risk factors (ultrasound patterns) of hospitalization, is a very useful tool for decision-making in primary care regarding how to perform the follow-up of patients. In fact, to our knowledge, we have not found any publication in this regard. Unlike the study by Calvo-Cebrian et al. [18] the presence of a diffuse interstitial pattern should not lead to hospital referral, but rather to greater patient control by primary care teams instead.

We also consider that, in the absence of other clinical criteria, the decision for hospital referral should not only be made on the basis of the LUS findings. In this sense, our study is in line with recommendations of Ma et al. [10] on the use of pulmonary point of care ultrasound (POCUS) to evaluate hospitalized patients with known or suspected COVID-19, above all with recommendation 6 [10].

### 4.4. Implications for Research and Practice

The capacity of LUS for early detection of COVID-19 pneumonia was already described in other publications [10,17,23]. Here, for the first time in primary care, we show the great relevance of LUS to predict the evolution of COVID-19 pneumonia diagnosed by ultrasound and confirmed by biological tests.

We found that different patterns of pathological LUS are risk factors and normal LUS is a protective factor for hospital admission. Moreover, during the acute phase of the disease, evolutionary patterns also represented a risk factor.

The results of this study have allowed us to design a follow-up method in our primary care centre. In patients with pulmonary patterns considered as a risk factor, we started treatment with enoxaparin in all cases and antibiotics and dexamethasone was considered appropriate. We carried out the first presential follow-up visit 24–72 h after the first and included acute phase blood reactants and D-Dimer, and radiological and/or ultrasound examination. The doctor who performed the follow-up visit decided whether to refer the patient to the hospital and, if not, when the next visit should take place during the following three weeks (in the most complicated cases) from the onset of symptoms. The decisions have been based on the clinical evaluation of the patient, the results of the blood test, and the ultrasound and/or radiological pattern. Patients with safe pulmonary patterns could undergo a less extensive follow-up.

This study shows that primary care services can and should be in charge of the management and follow-up of approximately 70% of patients with COVID-19 pneumonia. Despite this, many patients with mild symptoms prefer to go to hospital emergency departments such as the mild phenotype group (n = 402) of Volpicelli et al. [23]. These patients could be cared for by primary care services, decongesting hospital emergency services.

The usefulness of LUS has always been an element of debate in primary care, by the difficulty to transmit the echo by air and the presence of other exploratory techniques (chest X-ray or computer tomography). This study, in line with others [24], shows the great utility of this technique in primary care and also for the diagnosis of diseases such as COVID-19 pneumonia, lung cancer, interstitial pneumonia, and heart failure. In conclusion, we suggest that primary care services must continue with the research in the field of LUS and our study could represent a big step forward in this direction.

## 5. Conclusions

In our study we show that the risk of hospitalization in COVID-19 patients can be predicted by lung ultrasound in primary care, due to the presence of certain ultrasound patterns that act either as protective factors, or risk factors, for hospitalization. Therefore, lung ultrasound is very useful to establish how and when patients should be followed up by primary care services. According to our results, the decision to refer the patient to the hospital should not depend solely on the presence of a certain ultrasound pattern and should be taken after evaluating other clinical and analytical criteria.

## Figures and Tables

**Figure 1 ijerph-18-06083-f001:**
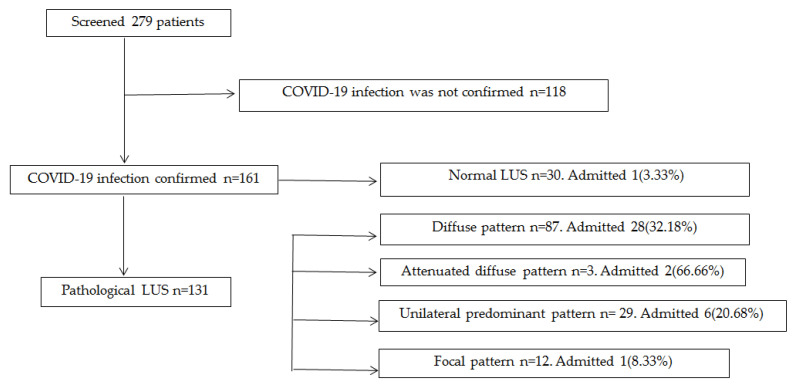
Sample distribution.

**Figure 2 ijerph-18-06083-f002:**
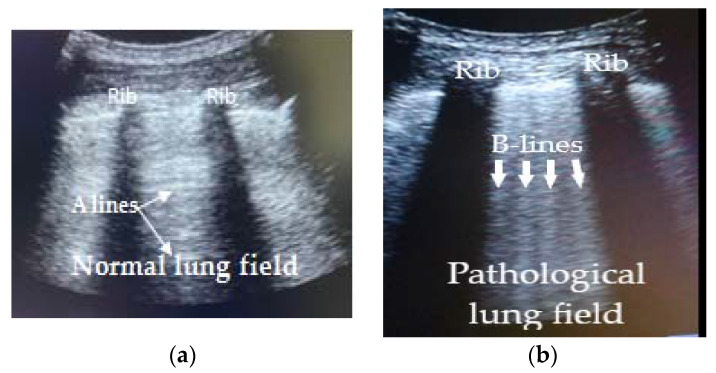
Difference between normal and pathological lung field in COVID-19 pneumonia. (**a**) Pulmonary field between two ribs. Normal pleural line with reverberation artefact (A Lines); (**b**) Pulmonary field between two ribs. Pleural thickening line with lung rockets artefacts (B Lines).

**Figure 3 ijerph-18-06083-f003:**
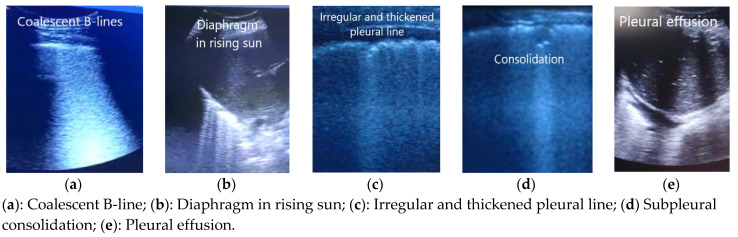
Others LUS findinds in patients with COVID-19 pneumonia.

**Table 1 ijerph-18-06083-t001:** Patterns description and number of cases.

Interstitial Pattern	Description	*n* = 172
Diffuse	Three or more B-lines in two or more lung fields in a longitudinal section between two ribs (bat sign) and bilaterally.	87
Attenuated diffuse	More than 8 B-lines distributed in an erratic bilateral inhomogeneous manner.	3
Unilateral predominant	Three or more B-lines in two or more lung fields in a longitudinal section between two ribs. This pattern was sometimes associated with isolated, erratically arranged B-lines (less than 8) or with a single contralateral field with 3 or more B-lines.	29
Focal	Three or more B-lines in a single lung field in a longitudinal section between two ribs and unilaterally, sometimes with isolated and associated B-lines (less than 8).	12
Normal	Absence of B-lines or, in some cases, B-lines (less than 8 and no more than 2 in a single lung field) erratically scattered.	30

**Table 2 ijerph-18-06083-t002:** Distribution of other ultrasound findings according to the different patterns.

LUS Findings	Diffuse Interstitial Pattern *n = 87*	Attenuated Diffuse Interstitial Pattern *n = 3*	Unilateral Predominant Interstitial Pattern *n = 29*	Focal Interstitial pattern *n = 12*
Coalescence of B-lines	15	0	1	2
Diaphragm in rising sun	6	0	0	0
Irregular and/or thickened pleura	20	0	1	0
Decreased sliding	2	0	0	0
Consolidations	21	0	0	0
Pleural Effusion	3	0	0	0

**Table 3 ijerph-18-06083-t003:** Distribution and association between hospital admissions and the different pathological LUS patterns.

	Diffuse Interstitial Pattern *n* = 87	Attenuated Diffuse Interstitial Pattern *n* = 3	Unilateral Predominant Interstitial Pattern *n* = 29	Focal Interstitial Pattern *n* = 12
Distribution of hospital admission according LUS pattern	28(32.18%)	2(66.66%)	6(20.68%)	1(8.33%)
Association between hospital admission and LUS pattern	*p* = 0.0016	*p* = 0.0002	*p* = 0.039	*p* = 0.4918

## Data Availability

This study did not report any data.

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
