# Peer review of "The Risk of Hospitalization in COVID-19 Patients Can Be Predicted by Lung Ultrasound in Primary Care"

_ijerph, 2021, doi:10.3390/ijerph18116083_

Round 1

Reviewer 1 Report

In the exclusion criteria you did not mention excluding prior lung disease, congestive heart failure, end stage lung disease. So how do we know that some of these patterns seen were not from prior lung disease, congestive heart failure or fluid overload from end stage renal disease. 

Table 3 needs to be worded differently

It is very confusing to see the n=172 and then attenuated diffuse pattern n=2 having a 66% rate

You should add all the n's of the various patterns in each category and then show the number of hospital admissions in each

According to the prior studies they d not recommend using lung ultrasound in COVID as the deciding factor for hospitalizations. Yet the authors have stated that this maybe the case.

Also could the authors clarify the symptoms present and the oxygenation in the cohort of hospitalized patients with abnormal lung ultrasound.

Could the authors also have representative images of the various patterns seen. 

Author Response

Dear Mr/Ms,

I attached the document which reply your suggestions.

Yours faithfully,

Marta Ortega

Reviewer 2 Report

The Covid-19 pandemic has placed great strain on various health care services in different parts of the world. Doctors have shown initiative in coping with these demands and a shortage of resources by using what tools they have in innovative ways. It id thus pleasing to see that the primary health care providers in Balaguer Spain have taken the initiative to not only use thoracic ultrasound to assess patients with Covid-19, but to take the trouble to publish their findings.

Unfortunately the data has been collected retrospectively rather than prospectively, which introduces the potential for biases in data collection.

It is not clear to me whether the ultrasound findings were used in making the decision about whether a patient should be admitted to hospital.

As the purpose of this review is to see if the ultrasound findings can predict who needs to go to hospital it would be most useful to have chi-square tables giving the numbers of those with or without different ultrasound findings who were and were not admitted to hospital. The sensitivities and specificities of the finds could then be calculated.

Also as it is retrospective it was not possible to assess the reliability and validity of the reporting of the ultrasounds.

A strength of the study is that all patients who presented with symptoms to suggest a chest infection had a chest ultrasound.  This then helps to determine the specificity of the ultrasound changes for Covid-19 lung involvement.

Author Response

(The authors gave the same response as above.)

Reviewer 3 Report

The use of LU in the evaluation of COVID-19 patients is encouraged because of its characteristics (e.g. low cost, radiation free). Several studies evaluated its use and it has been shown as a useful tool to evaluate progression, therapeutic response and follow-up of pulmonary disease in the patients with COVID-19.

This is a retrospective and descriptive study based on a small population (the final number of patients with LUS examination and a positive RT-PCR result was 161).

The conclusions are not particularly new or intriguing.

Some suggestions and questions:

  • line 38: please update the epidemiological information
  • line 235: all ultrasound examinations were performed by the same examiner; I think that this is not a strength because it reduce the reproducibility of these findings
  • table 3: 172 is the number of patients with pathological LUS but it includes also covid-19 negative patients; I think that considering also non covid-19 patients is confusing and not focused on the topic of the article
  • main data on table 3 are already reported on fjgure 1
  • line 214-216 I think that these data are not so interesting to introduce the conclusions

Author Response

(The authors gave the same response as above.)
